# Mesenchymal Stem Cells Delivered Locally to Ischemia-Reperfused Kidneys via Injectable Hyaluronic Acid Hydrogels Decrease Extracellular Matrix Remodeling 1 Month after Injury in Male Mice

**DOI:** 10.3390/cells12131771

**Published:** 2023-07-04

**Authors:** Daniel S. Han, Christopher Erickson, Kirk C. Hansen, Lara Kirkbride-Romeo, Zhibin He, Christopher B. Rodell, Danielle E. Soranno

**Affiliations:** 1Pediatric Urology, Department of Urology, Stanford University School of Medicine, Palo Alto, CA 94305, USA; 2Department of Biochemistry and Molecular Genetics, University of Colorado, Aurora, CO 80045, USA; 3Department of Pediatrics, University of Colorado, Aurora, CO 80045, USA; 4School of Biomedical Engineering, Science and Health Systems, Drexel University, Philadelphia, PA 19104, USA; 5Division of Nephrology, Department of Pediatrics, Indiana University School of Medicine, Indianapolis, IN 46202, USA; 6Department of Bioengineering, Purdue University, West Lafayette, IN 47907, USA

**Keywords:** ischemia-reperfusion acute kidney injury, cell therapy, injectable hydrogels, mesenchymal stem cells, proteomics

## Abstract

The translation of stem cell therapies has been hindered by low cell survival and retention rates. Injectable hydrogels enable the site-specific delivery of therapeutic cargo, including cells, to overcome these challenges. We hypothesized that delivery of mesenchymal stem cells (MSC) via shear-thinning and injectable hyaluronic acid (HA) hydrogels would mitigate renal damage following ischemia-reperfusion acute kidney injury. Acute kidney injury (AKI) was induced in mice by bilateral or unilateral ischemia-reperfusion kidney injury. Three days later, mice were treated with MSCs either suspended in media injected intravenously via the tail vein, or injected under the capsule of the left kidney, or MSCs suspended in HA injected under the capsule of the left kidney. Serial measurements of serum and urine biomarkers of renal function and injury, as well as transcutaneous glomerular filtration rate (tGFR) were performed. In vivo optical imaging showed that MSCs localized to both kidneys in a sustained manner after bilateral ischemia and remained within the ipsilateral treated kidney after unilateral ischemic AKI. One month after injury, MSC/HA treatment significantly reduced urinary NGAL compared to controls; it did not significantly reduce markers of fibrosis compared to untreated controls. An analysis of kidney proteomes revealed decreased extracellular matrix remodeling and high overlap with sham proteomes in MSC/HA-treated animals. Hydrogel-assisted MSC delivery shows promise as a therapeutic treatment following acute kidney injury.

## 1. Introduction

Acute kidney injury (AKI) is a common and known risk factor for progression to chronic kidney disease (CKD) in both adults and children [1]. AKI causes injury or death to tubular epithelial cells, precipitating a proinflammatory response that ultimately leads to fibrosis [2]. In the acute phase of renal injury, tubular epithelial cells are able to regenerate, and patients are able to regain normal kidney function. However, with persistent and/or severe injury, this regenerative process can become dysfunctional and result in a maladaptive process that leads to kidney fibrosis and CKD [3,4]. Regardless of the etiology of AKI, the only current treatment option is supportive care. The pathophysiological timing and signaling that causes AKI to result in CKD, as well as the duration of a therapeutic window, remains unclear. Our established murine model of bilateral ischemic AKI (BiAKI) results in a significant increase in serum IL-6, urine neutrophil gelatiniase-associated lipocalin (NGAL), collagen type 3 deposition, renal fibrosis, and α-smooth muscle actin activity, with a persistent decrease in the glomerular filtration rate (GFR) after injury [5]. We have recently demonstrated a direct correlation between ischemia duration and the (1) reduction in glomerular filtration rate and (2) development of kidney fibrosis [6]. Mitigating the transition from AKI to CKD could profoundly impact and reduce the clinical burden of kidney disease.

Shear-thinning hydrogels are water-swollen polymer networks that can be preformed in a syringe, including with encapsulated pharmacological or cellular therapeutics, and used to improve local therapeutic delivery [7]. When sufficient shear stress is applied, the material undergoes a transition to liquid-like flow that allows for ease of injection through a syringe needle or catheter. Following injection, the hydrogels rapidly revert back to a solid state, thereby retaining encapsulated cargo at the delivery site. These biomaterials can be engineered to deliver a variety of therapeutic agents, such as cytokines, growth factors, or cells, in a sustained manner. We have previously utilized shear-thinning injectable hyaluronic acid (HA) hydrogels to deliver IL-10 and anti-TGF-β in AKI and CKD murine models [5,8]. Results demonstrated reductions in fibrosis in all study groups, including control groups receiving HA hydrogel alone. We have also demonstrated our ability to quantify and localize hyaluronic acid hydrogels in vivo after injection under the kidney capsule in mouse models of UUO and bilateral ischemia-reperfusion [5,9].

Mesenchymal stem cells (MSCs) are multipotent stem cells that can be harvested from mesodermal tissue sources such as bone marrow but are most readily available from adipose tissue [10]. MSCs have been studied in preclinical models of adult kidney disease and have shown the ability to engraft into damaged renal tissue and produce antiapoptotic and promitogenic factors, exerting a beneficial effect on tubular recovery and renal function [11,12]. Additionally, MSCs have been found to have tropism to damaged tissue and an ability to promote survival of damaged cells [10,13]. In a rat model of ischemic renal injury, MSCs that engrafted into damaged renal tissue produced antiapoptotic, promitogenic, and vasculotropic factors, exerting a beneficial effect on tubular damage and renal function [11,12]. There have been multiple clinical trials investigating MSCs in the treatment of kidney disease; however, the majority of these trials utilized MSCs injected intravenously. In this situation, MSCs can be sequestered in the spleen or liver as they circulate. Additionally, the literature suggests that the main mechanism of MSC therapy in kidney injury is via paracrine actions [11,14]. Thus, it can be hypothesized that direct delivery to, and retention of MSCs in the kidney itself, would improve the impact of MSC therapy in kidney injury.

While promising, cell injection therapies face a number of obstacles that have hindered their translation in mainstream clinical use for regenerative medicine. Primary obstacles include the mechanical disruption of cells during the injection process and an inability to retain cell therapeutics at the injury site. Shear-thinning hydrogels are a unique tool that simultaneously address both of these concerns. Shear-thinning hydrogels have been used as a medium for cell delivery, shown to improve cell viability via a reduction in shear stress and extensional flow during injection [15,16]. Shear-thinning HA hydrogels have been previously demonstrated to serve as a scaffold for MSC encapsulation with high viability [17], with the capacity to improve cell viability postinjection and thereby support local cell retention and engraftment [18,19,20]. Following injection, cells are also subject to biological insults from the native tissue. Notably, injection into ischemic, damaged tissue can impact cell viability and engraftment due to their hypoxic and inflamed microenvironment [21]. MSC engraftment rates in nonhuman primates following intravenous infusion have been reported to range from 0.1 to 2.7%, with cells distributed throughout the body [22]. Though, intravenous infusion via the renal hepatic artery is associated with improved retention rates of 10 to 15% in the kidney [23]. Hydrogel encapsulation is an effective means of protecting delivered cells from the host foreign body response and maintaining cell localization, thereby improving engraftment at the injury site [20,24].

The purpose of this study is fourfold: (1) Determine the direct effect of injecting MSCs encapsulated within our HA hydrogel under the kidney capsule on measured glomerular filtration rate; (2) Determine MSC localization after unilateral or bilateral AKI when delivered systemically or delivered unilaterally under the kidney capsule, either within media or within HA hydrogel; (3) Determine the effect of unilateral MSC/HA therapy on renal outcomes after bilateral ischemia-reperfusion injury; (4) Determine the effect of bilateral injury and treatment on the kidney proteome. We hypothesize that utilizing HA hydrogels to deliver MSCs locally to damaged renal tissue in an ischemia-reperfusion AKI animal model will improve renal functional and histologic outcomes. 

## 2. Materials and Methods

### 2.1. Animals

All animal care and procedures were approved by the Institutional Animal Care and Use Committee at the University of Colorado, and conformed to the National Institutes of Health Guide for the Care and Use of Laboratory Animals. Adult (6–7-weeks-old) wild-type BalbC male mice were acquired (Jackson Laboratories, Bar Harbor, ME, USA) and housed in standard conditions with po ad lib access to water and chow. Males were chosen because of their known susceptibility to ischemia-reperfusion AKI compared to females. AIN-76A Rodent diet was administered to all cohorts from Research Diets Inc. (New Brunswick, NJ, USA) which decreased background fluorescence with imaging. Because female sex protects against ischemic AKI, only male mice were included in this study.

### 2.2. Unilateral Nephrectomy for Assessment of the Effect of MSC/HA on tGFR in Healthy Kidneys

We first sought to study whether subcapsular injection of MSC/HA into healthy kidneys adversely affected tGFR. These mice underwent a right unilateral nephrectomy and recovered for 28 days as previously described [25]. On day 28, the left kidney was injected with MSC suspended in either HA or media underneath the kidney capsule, as described below. The control group underwent right nephrectomy but did not receive any left renal subcapsular injection. On day 27, prior to MSC delivery the transcutaneous glomerular filtration rate (tGFR) was measured in the remaining left solitary kidney as described below. We then measured tGFR at days 1 and 7 after MSC injection (days 29 and 35 status post unilateral nephrectomy).

### 2.3. Ischemia-Reperfusion AKI

Adult mice were anesthetized with ketamine/xylazine and given bupivacaine locally at the incision site. The mice were prepped with betadine and 100% ethanol. A transabdominal approach was used, utilizing a longitudinal midline incision. The bowel was swept away and the renal pedicle(s) were isolated. For the unilateral AKI (UiAKI) groups, only the left pedicle was dissected and clamped. For the bilateral AKI (BiAKI) groups, both the left and the right pedicles were dissected and clamped. The clamp duration was 25 min; body temperature was maintained throughout the procedure, consistent with models of warm ischemia as opposed to cold ischemia. Sham mice underwent dissection of both renal pedicles without clamping. Closure was performed in 2 layers—the fascial layer was closed with 4-0 Vicryl (Ethicon, Sommerville, NJ, USA) then the skin layer using 4-0 Sofsilk (Covidien, Mansfield, MA, USA). The mice were injected with 0.8 mg/kg Buprenorphine SR (Zoopharm, Laramine, WY, USA) subcutaneously and 750 μL warmed normal saline subcutaneously post op. A total of 750 μL normal saline was administered subcutaneously daily for 5 days postoperatively. 

### 2.4. Measurement of Transcutaneous Glomerular Filtration Rate (tGFR)

Serial transcutaneous glomerular filtration rates (tGFRs) were measured as previously described [6]. Briefly, FITC-sinistrin was administered intravenously via retro-orbital injection and the renal clearance was measured for 65 minutes per the manufacturer instructions (MediBeacon GMBH, Manheim, Germany). Measurements of tGFR were normalized for body weight. In the unilateral nephrectomy groups, tGFRs were measured at baseline and pre- and post-MSC/HA injection as described above. In the unilateral and bilateral ischemia-reperfusion AKI groups, tGFR measurements were performed at baseline, 3 days after AKI, 1 week after treatment and prior to sacrifice 28 days after AKI or sham procedure.

### 2.5. MSC Encapsulation in HA Hydrogel and Delivery

Mesenchymal stem cells (MSCs) isolated from male mice were obtained from Cell Biologics (Chicago, IL, USA, Cat# BALB-5043). Cells were cultured under standard conditions (37 °C, 5% CO_2_) in complete mesenchymal stem cell medium (Cell Biologics, Chicago, IL, USA, Cat#M5566). MSCs were subsequently harvested at the first passage for all treatments. Injectable guest–host HA hydrogels were synthesized and prepared according to our previously reported methods [5,8,26]. Briefly, hyaluronic acid (HA; 74 kDa molecular weight; Lifecore, Caska, MN, USA) was converted to the tetrabutyl ammonium salt by ion exchange and separately modified by either 6-(6-aminohexyl)amino-6-deoxy-β-cyclodextrin or 1-adamantane acetic acid, yielding CD-HA and Ad-HA, respectively. Both HA derivatives were obtained with an average degree of substitution from 20 to 25%. Hydrogels were prepared from stock solutions of the two components, combined in a one-to-one stoichiometric ratio to yield HA hydrogels at 3.5% *w*/*v*. For cell inclusion, MSCs were suspended in Ad-HA immediately prior to hydrogel formation at a final density of 1 × 10^5^ cells per 15 μL of HA hydrogel. Under anesthesia, the left kidney was accessed via a dorsal approach, and the MSC/HA suspension was injected under the left kidney capsule. Other groups included MSC (1 × 10^5^) suspended in 15 μL of Corning^®^ media 199 injected either via the tail vein or under the left kidney capsule via the same approach described above. All treatments were administered 3 days after UiAKI or BiAKI.

### 2.6. Serial In Vivo Optical Imaging of Hydrogel and MSCs and Ex Vivo Organ Imaging

Serial in vivo optical imaging was performed to assess the localization of the HA hydrogel compound and the MSCs. MSCs were tagged with Qtracker^TM^ (Invitrogen, Eugene, OR, USA, Cat#: Q25061MP), which enabled in vivo imaging of live cells, and HA hydrogels were tagged by covalently bonding the near-infrared marker Cy7.5 to the Ad-HA component of the HA hydrogel as previously described [8,9]. The animals were imaged utilizing the Pearl^®^ imaging system (Li-Cor, Lincoln, NE, USA) prior to injection, immediately postinjection and on days 3–7, 14, 21, and 28. Signal intensity measured in photons/pixel/second was analyzed (Pearl Impulse Software v2.0), normalized per animal to peak intensity, and averaged per cohort.

### 2.7. Biomarker Measurement

Serial blood draws were performed via a retro-orbital approach and utilized to measure blood urea nitrogen (BUN) (BioAssay Systems QuantiChrom™ Urea Assay Kit Cat: (DIUR-500)), serum creatinine (Cr) (Pointe Scientific Creatinine (Enzymatic) Reagent Kit (Cat: C7548-480)), and Cystatin C (ELISA, R&D Systems, Minneapolis, MN, USA (Cat #:MSCTC0)). Additionally, urine was collected to measure KIM-1 and NGAL via ELISA (R&D Systems, Minneapolis, MN, USA (Cat #:MKM100 and MLCN20)).

### 2.8. Histological Quantification of Kidney Fibrosis

Mice were euthanized with phenobarbital, and the kidneys, spleen, and liver were harvested. The kidneys were cut longitudinally in half and one half was flash-frozen for subsequent proteomic analysis. The other half of the kidney was fixed in formalin, paraffin embedded, and cut into 4 μm sections. Histological samples were analyzed for fibrosis via Picrosirius Red (PSR), immunohistochemistry for collagen type 3, and hydroxyproline was measured as previously described [6].

### 2.9. Proteomics

Five kidneys from each sham, AKI, and treatment groups were processed for LC-MS/MS as previously described [27]. Briefly, the kidneys were lyophilized, delipidated with acetone, and proteins were extracted using a series of 3 M NaCl, 6 M Gnd-HCl, and Gnd-HCl with 1 M NH_2_OH-HCl (Sigma 431362) to target cellular, soluble extracellular (ECM), and insoluble ECM proteins. Samples were reduced (DTT), alkylated (iodoacetamide), trypsin digested on a FASP column, and desalted on a C18 resin tip. Samples were then dried, resuspended in 0.1% FA, and loaded onto an EASY n-LC II system coupled to the LTQ Orbitrap Velos Pro mass spectrometer (Thermo Scientific, CA, USA). Peptides were identified from the resulting mass spectra using MSFragger with the *Mus musculus* Uniprot reference proteome database (UP000000589. 1-25-2022). Variable (methionine oxidation, asparagine, and glutamine deamidation) and fixed (cysteine carbamidomethylation) modifications were taken into account. Cumulative peptide intensity values were compared across samples using R statistical software (version 4.2.3), and protein expression levels were determined to be statistically significant among groups via ANOVA with Tukey HSD post hoc or Student’s *t*-test and after Benjamini–Hochberg FDR adjustment (*q* < 0.1). Pathway enrichment was performed using Metascape [28]. Correlations between proteins and functional outcomes were performed using Spearman correlation, and correlations with *p*-value < 0.05 were considered significant.

### 2.10. Statistical Analysis

Data are reported as mean ± standard deviation (SD), unless otherwise indicated. Apart from the proteomics analysis described above, ANOVA with Tukey post hoc analysis was used for comparisons amongst 3 groups, with α < 0.05. Unpaired *t*-tests assuming Gaussian distribution with Welch’s correction were used for comparison between two cohorts, with α < 0.05. 

## 3. Results

### 3.1. The Effect of MSC/HA Injection on Measured Glomerular Filtration Rate in Healthy Kidneys

We first sought to assess the functional biocompatibility of MSC/HA to the kidney by administering subcapsular injections of MSC/HA to healthy kidneys. Because the MSC/HA therapy is unilateral (delivered only to the left kidney) we first established a solitary kidney model by performing a unilateral right nephrectomy and allowing the remaining left kidney one month to recover. Following recovery, we injected MSC/HA under the left kidney capsule and measured tGFR pre- and post-treatment. Figure 1 shows measured renal function via tGFR before and after the delivery of MSCs in healthy solitary kidneys. There was no difference in tGFR amongst the cohorts prior to subcapsular injection. One day after injection, on Day 29 after unilateral nephrectomy, cohorts that received MSCs via HA hydrogel (HA^MSC^) had a decrease in tGFR compared to cohorts that received MSCs via saline (Saline^MSC^); there was no significant difference between the MSC/HA cohort and controls. One week after injection, there was no difference in tGFR amongst the cohorts indicating the alteration in tGFR was transient.

After determining the functional biocompatibility of MSC/HA treatment to the kidney, we next sought to determine the efficacy of MSC/HA treatment in ischemia-reperfusion models of acute kidney injury. All correlating data in the sham control and sham treatment groups are reported in the Appendix A.

### 3.2. Localization of MSCs and HA

Both MSCs and HA were fluorescently labeled and in vivo optical imaging was utilized to quantify their localization after injection under the left kidney capsule. We quantified HA and MSC localization in both the treated (left) and untreated (right) kidneys to determine recruitment of both MSC and HA to the contralateral untreated kidney. Figure 2 shows the imaging intensity of HA and MSCs in the left and right kidneys after injection under the left kidney capsule. HA injected under the left kidney capsule showed clearance by 14 days post-injection for sham, UiAKI, and BiAKI mice (Figure 2A). There were background levels of HA hydrogel in the right kidney in UiAKI and BiAKI mice, with a significant increase in imaging intensity in sham mice 1 and 2 days after injection (day 4 and day 5) (Figure 2B). MSCs tracked to both kidneys following BiAKI, although the overall signal intensity remained higher in the left, treated kidney; MSCs stayed localized to the left kidney after UiAKI (Figure 2C,D). Mice treated with MSCs in media either via IV or the subcapsular route did not display imaging intensity within the kidneys above background levels. 

### 3.3. Kidney Function and Biomarker Assessments at 1 Month after BiAKI

One month after treatment following bilateral ischemia-reperfusion injury, there was no significant difference in tGFR among the AKI, AKI intravenous MSC (AKI^ivMSC^), and AKI subcapsular MSC/HA (AKI^MSC/HA^) groups (Figure 3A). tGFR was decreased for AKI mice treated with MSC/media (AKI^MSC/media^) compared to MSC/HA treatment. Urinary NGAL was decreased for all treatment groups that received MSC treatment compared to no treatment (Figure 3B). Blood urea nitrogen (BUN) was significantly elevated in the AKI^MSC/media^ group compared to AKI^ivMSC^ and AKI^MSC/HA^ groups (Figure 3C). There was no difference in serum creatinine (Figure 3D) or Cystatin C (Figure 3E) amongst all groups.

### 3.4. Tissue Histology and Assessment of Fibrosis after BiAKI

We evaluated the effect of HA and MSC treatment on mitigating renal inflammation and fibrosis 1 month after BiAKI. We assessed the effect of treatment on treated (left) and untreated (right) kidneys separately to determine the direct effect of these therapeutics on markers of kidney fibrosis (Figure 4). Quantification of Picrosirius Red staining showed an increase in fibrosis of the both the left and right kidneys in the MSC/media group compared to all other treatment groups; fibrosis was higher in this group compared to the untreated AKI controls in the left but not the right kidney (Figure 4A,B), including the untreated AKI controls. There was no significant difference in collagen 3 immunohistochemistry among the treated and untreated groups in either kidney; however, there was a decrease in Collagen 3 staining in the left kidney of the intravenousMSC-treated group compared to untreated controls (Figure 4C,D). Hydroxyproline content was increased in the subcapsular MSC/media group compared to the MSC/HA group and controls in both kidneys, and AKI^ivMSC^ was increased in the left kidney (Figure 4E,F). There was no significant reduction in hydroxyproline content in the MSC/HA group compared to untreated controls.

### 3.5. Proteome-Wide Changes following MSC Treatment of AKI

Analysis of the top 50 differentially expressed proteins (DEPs) revealed three distinct groups 1 month postinjury (Figure 5A and Appendix A). The sham group had a relatively high expression of proteins involved in fatty acid oxidation (FAO: CROT, ACOT8, and MLYCD) and mitochondrial proteins (NEDD4L and ECSIT) which were decreased in all injured groups. ‘Not recovered’ kidneys (consisting of AKI controls and kidneys from all treated groups) had high expression of fibrinogens (FGA, FGB, and FGG), histones (H1s), plasma proteins (TTR, ALB, AMBP, and APOA4), fibrillar collagens, and proteins involved in ECM organization (MATN2, ICAM1, AMBP, TOR1, FBN1, and FN1) as well as uromodulin (Tamm–Horsfall protein). The remaining ‘recovered’ group had reduced expression of these proteins compared to the ‘not recovered’ group. 

When comparing differences among treated groups only, there were no FDR-adjusted significant differences (Appendix A). However, using raw *p*-value < 0.05, analysis of the top 50 DEPs among treated groups revealed unique proteomic activity across a wide range of processes. In general, AKI^ivMSC^ had high levels of metabolic enzymes, red blood cell proteins, and proteins involved in translation that collectively enriched for catabolic processes; the AKI^MSC/Media^ group had high expressions of histones and keratins; while the AKI^MSC/HA^ group had low expression of these proteins (Appendix A). When plotting the proteomic expression of each treatment group vs. sham along their principal components, the sham and AKI^MSC/HA^ groups showed the highest overlap of kidney proteomes when compared to the other treatments. (Figure 5B). To further understand these differences, volcano analysis was performed to compare proteins expressed at significantly higher levels with a > two-fold change among treatment groups vs. sham (Appendix A). Inflammation, wound healing, and coagulation were enriched in the AKI^ivMSC^ and AKI^MSC/Media^ treatment groups; meanwhile, AKI^MSC/HA^ kidneys did not enrich for these pathways and also had lower levels of ECM and fiber organization (Appendix A).

To identify proteins associated with recovery outcomes, proteome-wide correlation analysis was performed, with spearman rho values > |~0.4| also being significant (Appendix A). BUN, serum creatinine, and cystatin C correlated positively with proteins involved in ECM organization, glycolytic enzymes (DLAT, PKLR), and clotting (PLG, F2) proteins (Figure 5C). Transcutaneous GFR correlated negatively with these proteins and positively with red blood cell proteins (ANK1, EPB41). Urine NGAL had high correlations with proteins involved in FAO and proteins involved in transcription, translation, and biosynthesis (NUP43, RPL10, DAZAP1). Differences between 24 h and 28-day BUN and creatinine correlations were noted. For creatinine, there was higher correlation with ECM proteins at 24 h, and higher correlations with translational proteins and proteases (DCPS, CTSZ) at 28 days. For BUN, there was higher correlation with serum proteins and clotting factors at 24 h, and higher correlations with proteins involved with FAO at 28 days. Finally, enrichment analysis was performed identify pathways upregulated in kidneys with improved function (Appendix A). Kidneys with decreased 28-day BUN, serum creatinine, and urine NGAL and increased tGFR had higher levels of proteins that enriched for cell adhesion, morphogenesis, and differentiation and biosynthetic processes and had lower levels of proteins involved in apoptosis, phagocytosis, degranulation, and ECM production and degradation (Appendix A).

## 4. Discussion

We have previously demonstrated the effect of the local delivery of IL-10 via HA hydrogels in a bilateral ischemia-reperfusion murine model of AKI in decreasing systemic inflammation and renal fibrosis [5]. We have also demonstrated that HA hydrogel treatment alone, even in the absence of IL-10, reduced markers of inflammation and renal fibrosis, and that HA hydrogel delivery under the kidney capsule does not impact tGFR [25]. The current study builds on our prior work by studying the effect of combination MSC/HA therapy in modulating renal damage from both unilateral and bilateral ischemia-reperfusion kidney injury.

Our results show that MSC/HA therapy resulted in MSC-improved localization to the injured kidneys compared to other treatment groups, decreased urine NGAL compared to untreated controls, and had improved outcomes of tGFR, BUN, and fibrosis compared to other treatment groups after BiAKI. These results differ from our prior study assessing IL-10/HA treatment in a mouse model of ischemic AKI that showed no decrease in NGAL after treatment. Additionally, we also demonstrated the biocompatibility of MSC/HA on measured GFR, which had no long-term untoward effects in healthy kidney function, similar to our previous study demonstrating that HA alone did not adversely affect the glomerular filtration rate [25].

### 4.1. Clinical Signifance of Urine NGAL

Neutrophil gelatinase-associated lipocalin (NGAL) is an acute phase protein secreted by many human tissue types [29]. With regards to kidney injury, NGAL is expressed in the early phase of AKI, appearing in urine 2 h after ischemic injury [30,31]. Its expression has been shown to be correlated with the duration and dose of ischemia. Increased NGAL has been shown to be a predictor for the initiation of dialysis and in-hospital mortality in both pediatric and adult patients [32]. NGAL has also been shown to be directly correlated with the magnitude of renal impairment in patients with CKD, and an independent predictor of CKD progression [33]. Various mechanisms have been proposed as to the role of NGAL in the acute phase of renal injury in the development of renal fibrosis. In a murine model of renal injury from ischemic AKI and bilateral nephrectomy, Skrypnyk et al. showed that the IL-6-mediated hepatocyte expression of NGAL is the primary source of NGAL from AKI [34]. They also showed that increased urinary NGAL from AKI appears to primarily result from the impaired proximal tubular reabsorption of NGAL. Other studies have shown that NGAL plays a major role in renal fibrosis, renal deterioration, and progression to CKD [35,36,37]. In our study, we observed decreases in both urinary NGAL and markers of renal fibrosis in mice subjected to BiAKI that were treated with MSC/HA. Although we did not see improvement in Cystatin C, creatinine, or tGFR in these mice, we observed a significant improvement in the reduction of fibrosis compared to other treatment groups. Further studies are needed to investigate the pathway for decreased NGAL observed in these mice—whether this is due to sequestration of NGAL by MSC/HA or if there is another underlying pathway where MSC/HA is downregulating NGAL expression. 

### 4.2. Markers of Inflammation

In our previous work, we demonstrated that local subcapsular delivery of either HA hydrogel alone, HA containing IL-10, or IL-10 in saline improved markers of renal fibrosis after AKI [5]. Similarly, we also previously demonstrated that the delivery of either anti-TGF-β or IL-10 within HA improved renal fibrosis in a murine model of CKD, while in combination their delivery worsened renal fibrosis [8]. The current study shows that, in the setting of BiAKI, both the treated (left) and untreated (right) kidneys had a reduction in kidney fibrosis markers (Picrosirius red and hydroxyproline) when treated with MSC/HA under the left kidney capsule compared to other treatment groups but not compared to the untreated controls. This corroborates with other studies that have shown decreases in renal fibrosis after AKI from MSC treatment. In a rat model of IR AKI, Chen et al. showed that MSC treatment alone reduced the expression of oxidative stress, inflammatory and apoptotic biomarkers, as well as increase antioxidative, anti-inflammatory, and antiapoptotic mediators [38].

### 4.3. Sequestration of MSCs Is Dependent on the Laterality of Ischemia-Reperfusion Injury

One notable result is that in mice subjected to BiAKI there was significantly higher sequestration of MSCs to the contralateral untreated kidney compared to mice subjected to UiAKI. This gives further credence to the tropic nature of these therapeutics to sites of renal injury. MSC tropism to sites of injury has been previously well described [10,13]. Interestingly, in the left kidney there was sustained signal intensity of MSCs in the UiAKI group but not in the BiAKI group. One explanation for this is that the tropism of MSCs to the contralateral kidney caused a decrease in MSC in the treated kidney. A possible future direction would be to study whether an increased dose of MSC in BiAKI would allow for a higher sustained signal intensity of MSC in the treated kidney.

### 4.4. Proteomics

The various treatment groups resulted in three main kidney proteomes: (1) Kidneys that were never injured, (2) Injured kidneys that recovered, (3) Injured kidneys that did not recover. ‘Not recovered’ kidneys had similar protein expression as AKI controls, namely high expressions of fibrinogens, histones, fibrillar collagens, proteins involved in ECM organization, and uromodulin. The remaining ‘recovered’ group had reduced expressions of these proteins, indicating decreased fibrosis and tissue damage in these animals at a proteomic level. While there were no FDR-corrected significant differences among treatment groups, MSC/HA kidneys had lower levels of proteins involved in ECM organization, catabolism, inflammation, and degranulation, suggesting that kidneys treated with MSC/HA had a progression in wound healing compared to kidneys subjected to other treatments. Additionally, MSC/HA kidney proteomes showed the highest overlap with sham. Overall, these data suggest a progression towards a sham proteome in MSC/HA kidneys compared to other treatment groups.

### 4.5. Strengths/Limitations and Future Directions

One strength of our study was in vivo tracking of both hydrogels and MSCs to determine the localization after treatment. We were able to determine that MSCs travel to untreated kidneys after bilateral injury. A future direction would be to determine if MSC/HA localizes to injured kidneys when injected subcutaneously, as this route of administration would be a much more feasible method for translational therapies in human patients with AKI. A second strength is the fact that kidney function was measured via tGFR in addition to serum and urine biomarkers. Other strengths include our comparison of unilateral vs. bilateral ischemia and of IV vs. subcapsular delivery of therapeutics and control groups that received the MSC/HA subcapsular, MSC IV, and MSC in media subcapsular treatments following sham surgery (Appendix A).

Our study also has limitations. One limitation of our study is that we only used male mice in our study. However, we previously showed that there are no major differences in outcomes with regards to sex as a biological variable in HA delivery to the kidney [25]. It is well known that female sex has a protective effect against ischemic AKI, which introduces important sex biases that were beyond the scope of the current study [39]. Indeed, we recently demonstrated significant sex biases in cardiorenal outcomes in a model of matched AKI between male and female mice [40]. We, therefore, chose to study one sex and selected males given their known propensity for kidney injury after ischemia-reperfusion AKI. We also specified that the MSCs delivered to the male mice were obtained from males so as to avoid any potential sex bias from cell lineage. Future studies should investigate whether the MSC/HA therapy has a similar effect in female mice, as well as if the sex of the donor MSCs and the recipients impact outcomes. Secondly, the UiAKI cohort in our model does not receive the same degree of kidney injury as BiAKI. Future directions could investigate the impact of treatment for UiAKI in conjunction with a contralateral nephrectomy. However, this is beyond the scope of the current work, where we wanted to prioritize differential MSC tracking in the setting of unilateral delivery in either unilateral or bilateral ischemia-reperfusion. Finally, we did not include a control group in this study that received HA hydrogel alone. We have previously demonstrated that that HA alone reduces inflammation and improves outcomes after ischemia-reperfusion AKI as well as in a unilateral ureteral obstruction model of chronic kidney disease [5,8]. However, the focus of the current study was to determine whether the route and mode of MSC delivery impacted outcomes. Therefore, whether the MSC/HA treatment group had improved outcomes over HA alone cannot be discerned from the current investigation. 

## 5. Conclusions

Our work shows that unilateral MSC/HA delivery under the kidney capsule improves cell localization to the injured tissue and decreases urinary NGAL and markers of renal fibrosis in both kidneys in mice subjected to bilateral ischemia-reperfusion injury compared to delivering MSCs suspended in media either intravenously or directly under the kidney capsule. Furthermore, proteomic analysis revealed decreases in protein expression related to inflammatory and fibrosis markers in ‘recovered’ kidneys subjected to MSC/HA therapy. MSC/HA treatment has promise as a potential translational therapeutic for the treatment of AKI. Further studies are needed to determine the optimal dosing and timing of therapeutic intervention, as well as route of administration.

## Figures and Tables

**Figure 1 cells-12-01771-f001:**
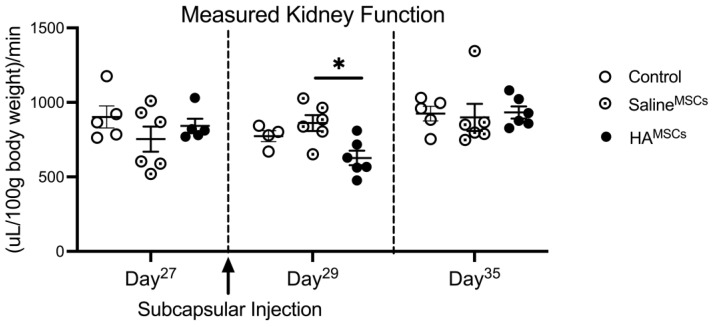
Legend: Measured tGFR after delivery of MSCs under the kidney capsule in healthy solitary kidneys. tGFR was measured in the remaining left solitary kidney 27 days after performing a unilateral right nephrectomy. On Day 28, MSCs were delivered under the left kidney capsule either via saline injection (Saline^MSC^) or encapsulated within HA hydrogel (HA^MSC^). There was no difference in tGFR amongst the cohorts prior to subcapsular injection. One day after injection, on Day29, cohorts that received MSCs via HA hydrogel had a decrease in tGFR compared to cohorts that received MSCs via saline; there was no significant difference between the HA^MSC^ cohort and controls. One week after injection, there was no difference in tGFR amongst the cohorts. n = 5–6, * indicates *p* < 0.05.

**Figure 2 cells-12-01771-f002:**
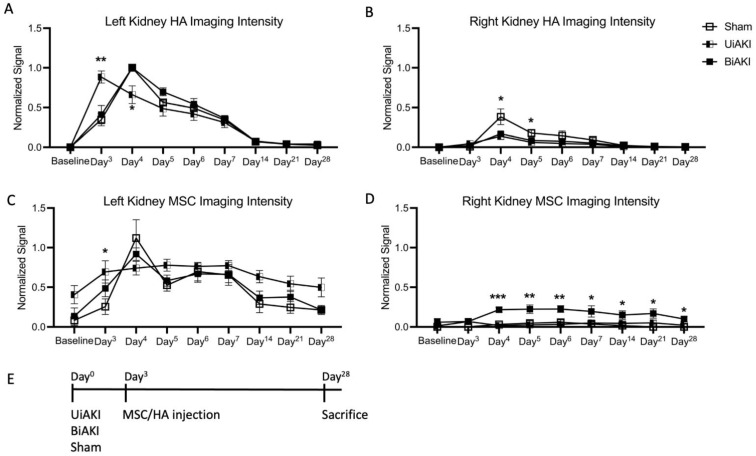
Legend: Serial in vivo optical imaging quantification of HA hydrogel in the left (**A**) and right (**B**) kidneys after delivery under the left kidney capsule 3 days after sham, unilateral left ischemia-reperfusion AKI (UiAKI) or bilateral ischemia reperfusion AKI (BiAKI); serial MSC quantification in the corresponding groups in the left (**C**) and right (**D**) kidneys. BiAKI resulted in bilateral locations of MSCs, whereas MSCs remained in the left kidney following UiAKI injury. (**E**) Schematic timeline of the experiment with either UiAKI, BiAKI, or sham procedure on day 0 and injection of MSC/HA under the kidney capsule on day 3. * indicates *p* < 0.05, ** indicates *p* < 0.01, *** indicates *p* < 0.001.

**Figure 3 cells-12-01771-f003:**
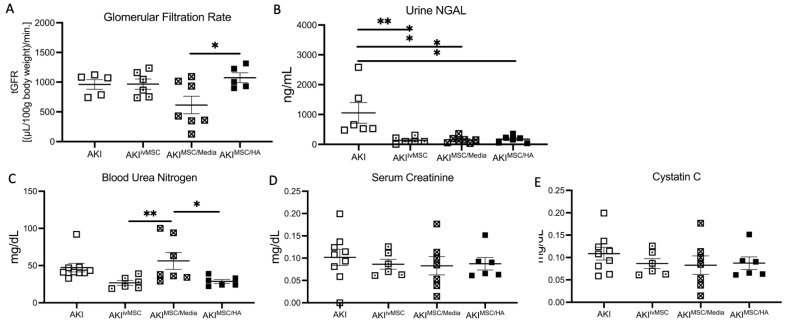
Legend: Renal functional outcomes at sacrifice, 1 month after bilateral ischemia-reperfusion acute kidney injury. Groups received treatment of either MSCs via IV (AKI^MSCiv^), MSCs injected under the left kidney capsule (AKI^MSC/media^) or MSCs encapsulated in HA hydrogel delivered under the left kidney capsule (AKI^MSC/HA^) 3 days after AKI. Measured tGFR (**A**), urine NGAL (**B**), blood urea nitrogen (**C**), serum creatinine (**D**) and serum cystatin C (**E**). * indicates *p* < 0.05, ** indicates *p* < 0.01.

**Figure 4 cells-12-01771-f004:**
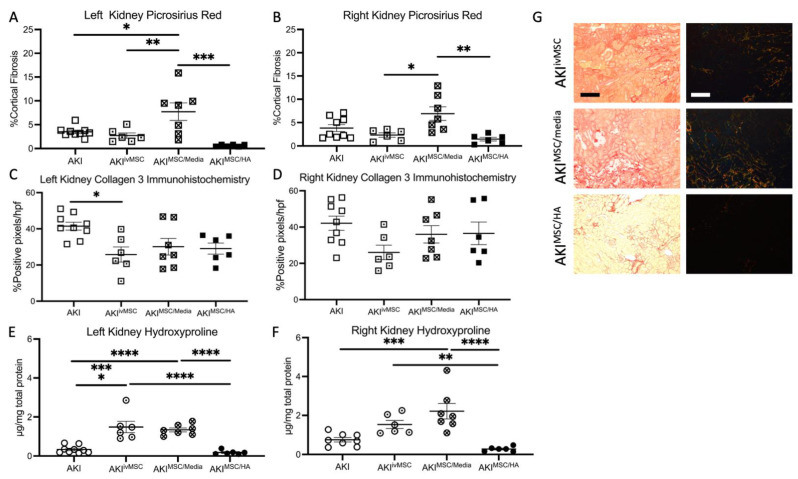
Legend: Kidney fibrosis outcomes at sacrifice, 1 month after bilateral ischemia-reperfusion acute kidney injury in the various treatment groups. Quantification of cortical fibrosis via polarized light after Picrosirius Red staining in the left (**A**) and right (**B**) kidneys. Quantification of immunohistochemistry staining for Collagen 3 in the left (**C**) and right (**D**) kidneys. Hydroxyproline content of the left (**E**) and right (**F**) kidneys. (**G**) Representative images of left kidneys from the MSC treatment groups, left panel light microscopy and right panel polarized light. Magnification 200×, scale bar = 10 μM. looOverall, groups treated with MSCs via HA hydrogel had improved fibrosis outcomes compared to the other delivery routes. * indicates *p* < 0.05, ** indicates *p* < 0.01, *** indicates *p* < 0.001, **** indicates *p* < 0.0001.

**Figure 5 cells-12-01771-f005:**
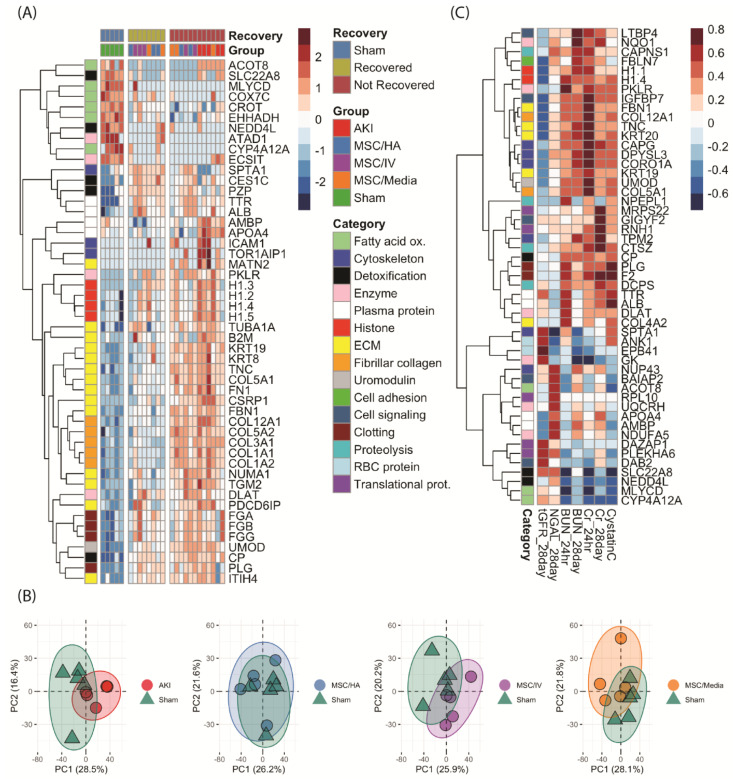
(**A**) Heatmap of sham, AKI, and MSC treatments, showing clustering into sham; ‘not recovered’ kidneys with high levels of ECM proteins, histones, and fibrinogens; and ‘recovered’ kidneys with reduced levels of these proteins. The y-axis groups proteins into general categories. (**B**) PCAs showing overlap of whole proteomes of sham (green) vs. AKI, MSC/HA, MSC/IV, and MSC/Media treated kidneys. (**C**) Heatmap showing correlation values and clustering of highly or lowly correlated proteins (y-axis) vs. functional outcomes (x-axis).

## Data Availability

Data supporting the findings of this study are available from the corresponding author (D.E.S.) upon reasonable request.

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
