# Peer review of "Mesenchymal Stem Cells Delivered Locally to Ischemia-Reperfused Kidneys via Injectable Hyaluronic Acid Hydrogels Decrease Extracellular Matrix Remodeling 1 Month after Injury in Male Mice"

_cells, 2023, doi:10.3390/cells12131771_

Round 1
Reviewer 1 Report
The manuscript investigates the delivery of mesenchymal stem cells (MSC) via shear-thinning and injectable hyaluronic acid (HA) hydrogels as a potential therapy to mitigate renal damage following ischemia-reperfusion acute kidney injury. The issue and rational are well contextualized. Nevertheless, the study plan presents several design flaws, lack of critical information, and to my opinion, the presented results do not support the proposed conclusions. Therefore, I could not support the publication of this manuscript in its current form.
In future resubmissions, the authors must address the following points:
1- Hyaluronic acid (HA) plays major roles in organogenesis and ECM remodeling. It was thoroughly studied during the last decades as either a biomaterial and a regulatory molecule. In particular, it was observed that HA can trigger paradoxical tissue responses, being pro- or anti- angiogenic, and pro- or anti-inflammatory, depending on its molecular weight. Moreover, the response of mesenchymal stem cells to HA matrices depend on the matrix stiffness, and availability of anchor sites. Cells can interact directly with HA matrix through extracellular proteins, such as CD44. Both the carboxylate oxygen’s of GlcUA7 and N-acetyl group of GlcNAc6 from HA disaccharide are important for CD44-HA binding. Herein is not presented any information regarding the HA molecular weight, modification degree, nor polymer concentration relevant for the assessment of the biological role of the HA hydrogel in the experimental outcomes. This information should be provided, at least as supplementary data.
2- Regarding the used MSCs, what was the cell passage and how was determined the adequate cell suspension density for the MSC/HA group? Moreover, it’s not clear from the presented data if the MSC survived to encapsulation and implantation. The authors should present sound evidences that the MSCs survived to the implantation procedure in order to test their hypothesis.
3- Further than the quantitative analysis, the authors should present histological evidences of the MSCs and HA presence in the tissues. Moreover, representative Picosirius Red stained (and/or H&E stained) histological sections should be presented for proper analysis of the tissues anatomy.
4- Two MSC topic implantation groups were studied: the MSC/HA and the MSC/medium. This last refered to “MSC (1x105) suspended in 15 uL of Corning® media 199”. Was the culture medium supplemented with FBS or any other animal serum? If so, how do the authors expect the mice immune system respond to xenogenic serum and how does it could affect the study outcomes?
5- How do the MSC/HA group outcomes would compare with the sole HA hydrogel injection? The HA itself has immunomodulatory properties, as reported by the authors in previous studies. It’s not clear the advantage of the encapsulation of MSCs in the HA hydrogels. The authors should justify the choice for not including a hydrogel control group and discuss properly the results that support a clear advantage of MCSs delivery with the HA hydrogels over the HA hydrogels alone.
6- The Figure 1 refers to a “SalineMSC” test group that is not described anywhere in the article.
Author Response
Reviewer 1:
The manuscript investigates the delivery of mesenchymal stem cells (MSC) via shear-thinning and injectable hyaluronic acid (HA) hydrogels as a potential therapy to mitigate renal damage following ischemia-reperfusion acute kidney injury. The issue and rational are well contextualized. Nevertheless, the study plan presents several design flaws, lack of critical information, and to my opinion, the presented results do not support the proposed conclusions. Therefore, I could not support the publication of this manuscript in its current form.
In future resubmissions, the authors must address the following points:
- Hyaluronic acid (HA) plays major roles in organogenesis and ECM remodeling. It was thoroughly studied during the last decades as either a biomaterial and a regulatory molecule. In particular, it was observed that HA can trigger paradoxical tissue responses, being pro- or anti- angiogenic, and pro- or anti-inflammatory, depending on its molecular weight. Moreover, the response of mesenchymal stem cells to HA matrices depend on the matrix stiffness, and availability of anchor sites. Cells can interact directly with HA matrix through extracellular proteins, such as CD44. Both the carboxylate oxygen’s of GlcUA7 and N-acetyl group of GlcNAc6 from HA disaccharide are important for CD44-HA binding. Herein is not presented any information regarding the HA molecular weight, modification degree, nor polymer concentration relevant for the assessment of the biological role of the HA hydrogel in the experimental outcomes. This information should be provided, at least as supplementary data.
The biological role of HA in the modulation of cell behavior, wound healing, and other tissue responses is well appreciated. The synthesis, characterization, and methods of hydrogel preparation for these materials have been previously reported in detail (Loebel & Rodell, Nature Protocols 2017). The manuscript has been updated to include necessary details pertinent to this study and interpretation of the results that include HA molecular weight (74kDa, sourced from Lifecore), an overview of synthesis procedures, and the degree of substitution.
2- Regarding the used MSCs, what was the cell passage and how was determined the adequate cell suspension density for the MSC/HA group? Moreover, it’s not clear from the presented data if the MSC survived to encapsulation and implantation. The authors should present sound evidences that the MSCs survived to the implantation procedure in order to test their hypothesis.
We thank the reviewer for these questions regarding MSC encapsulation and viability. We have clarified the process in the methods section and also added citations on our prior viability work with HA cell delivery upon which we based these methods. As noted above, protocols for hydrogel preparation (including for cell encapsulation) have been previously published in detail. These methods result in excellent viability of encapsulated MSCs (>95% at 24 hours; Rodell, Advanced Materials 2016) and have been demonstrated to improve cell viability during injection and thereby support local cell engraftment post-injection (Gaffey, The Journal of Thoracic and Cardiovascular Surgery 2015). MSCs were purchased from Cell Biologics as described in the methods. We ensured that all cells were obtained from male mice so as to avoid any potential sex biases in the results. All encapsulated cells were from the first passage. We used QTracker to label the MSCs for serial in vivo optical imaging and followed the manufacturer’s instructions for optimizing cell viability, incubating at 37C for 60 minutes. QTracker only fluoresces with live cells, such that our in vivo optical imaging indicated not only the localization of the cells, but the localization of live cells.
- Further than the quantitative analysis, the authors should present histological evidences of the MSCs and HA presence in the tissues. Moreover, representative Picosirius Red stained (and/or H&E stained) histological sections should be presented for proper analysis of the tissues anatomy.
We have added representative Picrosirius Red images from the left kidney MSC treatment groups, Figure 4G.
4- Two MSC topic implantation groups were studied: the MSC/HA and the MSC/medium. This last refered to “MSC (1x105) suspended in 15 uL of Corning® media 199”. Was the culture medium supplemented with FBS or any other animal serum? If so, how do the authors expect the mice immune system respond to xenogenic serum and how does it could affect the study outcomes?
After labeling with QTracker and then rinsing off the QTracker, the final suspension of MSC/Media199 was prepared for delivery. In HA groups, this suspension was encapsulated within the HA hydrogel during formulation. In non-HA delivery groups, the MSCs/Media199 were injected either under the kidney capsule or via tail vein injection. The media199 was not supplemented with FBS or other animal serum as the MSCs/media199 were promptly injected in vivo after final formulation. As above, the QTracker enables tracking of live cells, therefore, the optical imaging analysis demonstrates the viability of the cells under the preparation methods.
5- How do the MSC/HA group outcomes would compare with the sole HA hydrogel injection? The HA itself has immunomodulatory properties, as reported by the authors in previous studies. It’s not clear the advantage of the encapsulation of MSCs in the HA hydrogels. The authors should justify the choice for not including a hydrogel control group and discuss properly the results that support a clear advantage of MCSs delivery with the HA hydrogels over the HA hydrogels alone.
We thank the reviewer for this question and agree that HA alone has demonstrated a positive impact in kidney and systemic outcomes in our prior published work. For this study, we focused on the mode of MSC delivery. The aim was to determine whether HA/MSC delivery was superior to MSC delivery alone (either directly delivered to the kidney under the capsule, or via systemic venous injection). Therefore, the main groups of interest for this study were: MSC/HA, MSC/media199iv, MSC/media199sc. The other aspect of this experiment of main interest to us was whether the MSCs would differentially locate under the setting of unilateral versus bilateral ischemia-reperfusion injury. For these reasons, we chose not to repeat our prior HA only experiments, which we have previously published, but do reference them in our manuscript. Unfortunately, given the heterogeneity of the ischemia-reperfusion AKI model, we cannot compare the results from our prior works to the current study – we have found significant variability in the injury model with respect to seasonality and weather, and therefore all groups must undergo the experimental protocol concurrently. We have added this important limitation to our discussion and will include this additional control group in future studies.
6- The Figure 1 refers to a “SalineMSC” test group that is not described anywhere in the article.
Thank you. We have corrected this error in both the main text as well as the figure legend.
Reviewer 2 Report
The manuscript deals with a methodological improvement to make cell therapy more efficient. For this aim, mesenchymal stem cells were delivered to damaged kidneys (ischemia-reperfusion damage) packed within a hyaluronic acid (HA)-based hydrogel and delivered under the renal capsule. This delivery was compared to cells injected into the tail vein or under the kidney capsule but not in HA. To measure the repair capacity of the kidneys state-of-the-art functional tests and proteomic analysis were applied. As a result, MSC delivered by HA provided a promising new approach to enhance kidney repair capacity after damage.
The topic is relevant and important in the stem cell therapy field, the study was conducted in a scientifically sound way and the results were presented clearly and reproducibly.
However, only male mice were used as an experimental model. This was mentioned in the discussion as a limitation (p12 lines 446 ff). However, the reason given for omitting females was not convincingly explained. A protective effect of estrogen against ischemic AKI was mentioned: “It is well known that estrogen has a protective effect against ischemic AKI, which is why we excluded females from this study [36]”
This explanation, however, can easily be misunderstood to mean that AKI were not a problem for females at all, which is not true. Actually the damaging effect of ischemia was very much present in the female mice in the work of Hutchens et al [36]. The topic should be better elaborated and backed by more evidence. The implications of the mice studies for patients should also be discussed.
With reference to the work of the authors, it would be very straightforward to find out, if the HA delivered MSCs would be able to rescue kidney function in female mice similarly or potentially better than in males. This would open up possibilities to research the mechanisms underlying potential sex differences.
In addition, the SAGER guidelines https://ease.org.uk/communities/gender-policy-committee/the-sager-guidelines/ should be consulted. For example, title and abstract should address the fact that the study was conducted using male mice only.
Furthermore, the explanation in the Methods section for using males only should be rephrased and backed up by citations.
Minor comments
Some authors’ affiliations do not appear on the front page.
Inconsistencies in literature numberings (citation No 9)
Author Response
Reviewer 2:
The manuscript deals with a methodological improvement to make cell therapy more efficient. For this aim, mesenchymal stem cells were delivered to damaged kidneys (ischemia-reperfusion damage) packed within a hyaluronic acid (HA)-based hydrogel and delivered under the renal capsule. This delivery was compared to cells injected into the tail vein or under the kidney capsule but not in HA. To measure the repair capacity of the kidneys state-of-the-art functional tests and proteomic analysis were applied. As a result, MSC delivered by HA provided a promising new approach to enhance kidney repair capacity after damage.
The topic is relevant and important in the stem cell therapy field, the study was conducted in a scientifically sound way and the results were presented clearly and reproducibly.
However, only male mice were used as an experimental model. This was mentioned in the discussion as a limitation (p12 lines 446 ff). However, the reason given for omitting females was not convincingly explained. A protective effect of estrogen against ischemic AKI was mentioned: “It is well known that estrogen has a protective effect against ischemic AKI, which is why we excluded females from this study [36]”
This explanation, however, can easily be misunderstood to mean that AKI were not a problem for females at all, which is not true. Actually the damaging effect of ischemia was very much present in the female mice in the work of Hutchens et al [36]. The topic should be better elaborated and backed by more evidence. The implications of the mice studies for patients should also be discussed.
With reference to the work of the authors, it would be very straightforward to find out, if the HA delivered MSCs would be able to rescue kidney function in female mice similarly or potentially better than in males. This would open up possibilities to research the mechanisms underlying potential sex differences.
In addition, the SAGER guidelines https://ease.org.uk/communities/gender-policy-committee/the-sager-guidelines/ should be consulted. For example, title and abstract should address the fact that the study was conducted using male mice only.
Furthermore, the explanation in the Methods section for using males only should be rephrased and backed up by citations.
We thank the reviewer for their thoughtful comments regarding sex as a biological variable. We very much agree with the reviewer that both sexes need to be included in translational research. Indeed, this is an area of particular focus for our lab. Estrogen is protective against ischemia-reperfusion AKI, while testosterone is deleterious. Based on these known sex biases, we have recently developed a matched model of ischemia-reperfusion AKI in which females required 34 minutes of ischemia duration, compared to 25 minutes in males. These different ischemia models in males and females resulted in a matched degree of kidney injury based on tGFR measurements, out to 1 year after injury. Despite the matched kidney functional injury, this model resulted in differential cardiorenal outcomes between males and females (Soranno, et al. Scientific Reports 2022). Females had worse kidney fibrosis compared to males, but they maintained normal cardiac function while males developed diastolic dysfunction.
Because the focus of the enclosed study was to determine the impact of the route of MSC delivery on renal outcomes, and the fact that males and females require a different injury model (25 minutes vs 34 minutes of ischemia, respectively) that results in different renal histological outcomes, we focused this experiment on one sex, otherwise we would have needed to duplicate the entire study and use both the male and female AKI models (with different ischemia durations) which would have introduced substantially increases costs, animals, and more variables with respect to renal outcomes.
Males were chosen for this study over females, not because ischemia-reperfusion AKI is unimportant in females, but rather because: 1) the injury model requires less ischemia duration in males for the same degree of AKI based on tGFR measurements, 2) males are historically preferred over females for I/R because of their worse renal injury (whereas females are preferred for nephrotoxic models), 3) we were able to specify the sex of the MSCs and chose to match male MSCs to male mice.
Additionally, our prior published work demonstrated that our HA hydrogel does not impair kidney function in either males or females after delivery under the kidney capsule, so we felt reassured that the HA-aspect of this work would not be confounded by sex. Therefore, in this study that focused on the differential therapeutic efficacy of various modes of MSC delivery, we focused on one sex and chose males for the aforementioned reasons.
We agree that future investigations are warranted to query sex biases in a rigorous fashion, including: 1) delivery of female MSCs to female mice after AKI, 2) delivery of male MSCs to female mice, and 3) delivery of female MSCs to male mice.
We have added relevant citations and bolstered our discussion accordingly, highlighting this important issue in translational research. We have also specified the use of male mice in the title. Thank you for this constructive critique.
Minor comments
Some authors’ affiliations do not appear on the front page.
Thank you, we have corrected the affiliations to the front page.
Inconsistencies in literature numberings (citation No 9)
Thank you. We have corrected this formatting error.